# Nonlocal Reaction–Diffusion Model of Viral Evolution: Emergence of Virus Strains

**Nikolai Bessonov [1], Gennady Bocharov [2,3], Andreas Meyerhans [2,4,5], Vladimir Popov [6] and Vitaly Volpert [2,6,7,8,\***

[1]  Institute of Problems of Mechanical Engineering, Russian Academy of Sciences,
    199178 Saint Petersburg, Russia; nickbessonov1@gmail.com
[2]  Marchuk Institute of Numerical Mathematics, Russian Academy of Sciences, 199333 Moscow, Russia;
    bocharov@m.inm.ras.ru (G.B.); andreas.meyerhans@upf.edu (A.M.)
[3]  Key Center of Excellence on Experimental immunophysiology and immunochemistry,
    Ural Federal University, 620002 Ekaterinburg, Russia
[4]  Institució Catalana de Recerca i Estudis Avançats (ICREA), Pg. Lluis Companys 23, 08003 Barcelona, Spain
[5]  Infection Biology Laboratory, Universitat Pompeu Fabra, 08003 Barcelona, Spain
[6]  Peoples' Friendship University of Russia (RUDN University), 6 Miklukho-Maklaya St,
    117198 Moscow, Russia; popov-va@rudn.ru
[7]  Institut Camille Jordan, UMR 5208 CNRS, University Lyon 1, 69622 Villeurbanne, France
[8]  INRIA Team Dracula, INRIA Lyon La Doua, 69603 Villeurbanne, France
\*   Correspondence: volpert@math.univ-lyon1.fr

**Abstract:** This work is devoted to the investigation of virus quasi-species evolution and diversification due to mutations, competition for host cells, and cross-reactive immune responses. The model consists of a nonlocal reaction–diffusion equation for the virus density depending on the genotype considered to be a continuous variable and on time. This equation contains two integral terms corresponding to the nonlocal effects of virus interaction with host cells and with immune cells. In the model, a virus strain is represented by a localized solution concentrated around some given genotype. Emergence of new strains corresponds to a periodic wave propagating in the space of genotypes. The conditions of appearance of such waves and their dynamics are described.

**Keywords:** virus density distribution; genotype; virus infection; immune response; resistance to treatment; nonlocal interaction; quasi-species diversification

## 1. Introduction

Human infections with rapidly evolving viruses such as the human immunodeficiency virus (HIV) or the hepatitis C virus (HCV) remain a challenge for health-care systems. Infections are usually initiated by one or few virions that then replicate and generate a swarm of progeny viruses with distinct but related genomes [1,2]. Collectively these swarms of viruses are called a virus quasi-species [3–7]. This quasi-species nature enables viruses to rapidly evolve within an infected host organism and adapt to constraints mediated by immune responses or antiviral drugs [8,9]. It also allows viruses to broaden their host cell tropism and to spread to diverse tissues [10]. Well studied examples for virus adaptation are the development of drug resistance or the generation of variants within virus-specific cytotoxic T lymphocyte (CTL) epitopes that diminish immune recognition and destruction of infected cells [11–13]. Since the immune system can also adapt to respective virus changes [14], an increase in

the number of CTL target regions over time of infection as well as successive shifts in the hierarchy of immunodominance have been observed [15,16].

Gaining a mechanistic understanding of the dynamic interplay between the processes of virus replication, mutation, and elimination by immune responses and drug-based treatment requires the development of mathematical models which could be used to predict the generation of viral variants that escape the immune recognition and confer resistive to antiviral drugs. The existing models of virus evolution are based on the concept of quasi-species. i.e., an ensemble of related genomes [4]. The models can be formulated either as deterministic high-dimensional systems of ODEs, describing the densities of individual strain [15,17] or stochastic models with genetic algorithms [18]. The cooperative interactions in viral populations are considered to be key for linking the quasi-species dynamics in a changing virus-host environment with the genetic markers of viral evolution and the disease pathogenesis [10,19]. This implies that nonlocal interactions between the quasi-species in the genotype space need to be considered to predict the evolution of viruses to form distinct phenotypes.

Nonlocal reaction–diffusion equations represent an appropriate framework to describe evolution of biological species [20–22]. These equations take into account nonlocal consumption of resources characterizing intraspecific competition and possibly leading to the emergence of multi-modal population density distributions. Considered to be depending on a morphological characteristic and on time, localized in-space distributions can be interpreted as biological species, and the emergence of multi-modal distributions corresponds to the appearance of new species. In this work we will study virus quasi-species and will analyze the emergence of new strains in the space of genotypes. We consider the nonlocal reaction–diffusion equation

$$\frac{\partial u}{\partial t} = D\frac{\partial^2 u}{\partial x^2} + ru(1 - qJ(u)) - uf(S(u_\tau)) - \sigma(x)u \qquad (1)$$

introduced in the previous work [23] devoted to the existence and dynamics of virus strains, but not to the emergence of new strains since this question is different both from the biological and modelling points of view. Here $u(x,t)$ is a dimensionless virus density distribution depending on its genotype $x$ considered to be a continuous variable and on time $t$. The diffusion term in the right-hand side of this equation characterizes virus mutations, and the other terms describe virus reproduction, its elimination by immune response and by genotype-dependent mortality, either natural or caused by an antiviral treatment.

We describe the virus reproduction and immune response terms in more detail. We begin with the diffusion term. Assuming that there is a sequence of reversible mutations with consecutive genotypes $x_i$, we can write the equation for the density $u_i$ of virus with genotype $x_i$:

$$\frac{du_i}{dt} = \mu(u_{i-1} - u_i) + \mu(u_{i+1} - u_i),$$

where $\mu$ is the frequency of mutations. This equation represents a discretization of the diffusion equation with the diffusion coefficient proportional to $\mu$.

The virus reproduction rate is conventionally considered either proportionally to its density $u$ or, if we take into account the limitation on the quantity of the host cells where virus can multiplicate, as a logistic term $ru(1 - qu)$. Here $r$ is a proportionality coefficient, and $q$ is a positive constant corresponding to the inverse carrying capacity in population dynamics. The latter case implicitly signifies that there is one-to-one correspondence between the virus genotype and the type of infected cells. In a more general and biologically realistic setting we should accept that viruses with different genotypes can infect the same cells. In this case, we replace the conventional logistic term by the term $ru(1 - qJ(u))$, where $J(u) = \int_{-\infty}^{\infty} \phi(x - y)u(y,t)dy$. The kernel $\phi(x - y)$ in this integral characterizes the efficacy of host cell infection depending on the difference in genotypes. In general, it is a decreasing function of the modulus of its argument. Its exact form in the applications is not known, and different examples will be considered below. The integral is taken in the infinite limits for convenience of

presentation. It implies that the genotype space is sufficiently large, and it can be mathematically approximated as a real line.

The reproduction rate with the integral $J(u)$ corresponds to nonlocal consumption of resources in population dynamics [22,24]. If we replace the kernel $\phi(x)$ by the $\delta$-function, then we obtain the previous "local" case. Finally, if cell contamination is independent of virus genotype, then we have the integral $I(u) = \int_{-\infty}^{\infty} u(y,t)dy$ corresponding to the global consumption of resources. Behavior of solutions of Equation (1) can be essentially different in these three cases.

Virus elimination by immune cells is proportional to the virus density $u$ and to the concentration of immune cells $C$. Since immune response is stimulated by the antigen (virus), then the concentration of immune cells can be considered to be a function of virus density $C = f(u)$. The function $f(u)$ characterizes the intensity of immune response. It is positive and growing for some limited interval of values of $u$ where the antigen stimulates the immune response, and it is decreasing for $u$ sufficiently large due to the exhaustion and death of immune cells provoked by large virus concentrations [25–27]. The qualitative form of this function is well described by the dependence $f(u) = (k_1 u + k_2)e^{-k_3 u}$ with some positive constants $k_1, k_2$, and $k_3$. The approximation of the concentration of immune cells as a function of virus density can be derived from a more complete model under the assumption of large reaction rate constants [28].

Clonal expansion of immune cells requires several cell proliferations and differentiations, and it usually takes 3–4 days. Therefore, the rate of virus elimination by immune cells can take into account this time delay if it is not negligible in the time scale of virus evolution. In this case, the corresponding term becomes $uf(u_\tau)$, where $u_\tau = u(x, t - \tau)$. Furthermore, similar to virus reproduction, virus elimination is also nonlocal in the genotype space. In [23] it was described by the term $\int_{-\infty}^{\infty} \theta(x - y)f(S(u_\tau)(y,t))dy$, where $S(u_\tau)(y,t) = \int_{-\infty}^{\infty} \psi(y - z)u(z, t - \tau)dz$. The inner integral $S$ characterizes a cross-reactive stimulation of immune response by different antigens, while the outer integral describes a cross-reactive virus elimination by different immune cells. Both assumptions are biologically justified. However, this model becomes excessively complex, and we will restrict ourselves here only to the inner integral assuming the outer term is local, i.e., that the kernel $\theta(x)$ is replaced by the $\delta$-function.

The last term in the right-hand side of Equation (1) describes virus mortality with the rate depending on its genotype. The viability interval, i.e., the rate of genotypes where virus multiplication rate exceeds its mortality can depend on its intrinsic features and on an antiviral treatment.

Some particular cases of Equation (1) are studied in the literature. Considered without immune response and genotype-dependent mortality, this nonlocal reaction–diffusion equation and some its variations were widely studied in relation to various applications [29–32] and from the point of view of their mathematical properties [33–36]. One of the main features of this equation is that its homogeneous in-space stationary solution can become unstable leading to the emergence of periodic in-space solutions. We will return to this question below. The local equation ($J(u) \to u, S(u) \to u$) with time delay in the immune response term was suggested in [23,26,27,37] as a model of virus spread in tissues. The presence of time delay can lead to complex patterns of wave propagation.

In our previous work [23] we studied the existence and dynamics of virus strains considered to be localized solutions (pulses) in the space of genotypes, with the understanding that a virus strain can be characterized by its most frequent genotype with a narrow density distribution around it. The existence and stability of such solutions is not a priori given. In the local bistable equation such solutions exist but they are not stable. In the local monostable equation such solutions do not exist. It was previously shown that stable pulses exist for the nonlocal bistable equation [38]. In [23], it was revealed that persistent virus strains can exist due to the interaction of nonlocal (global) virus reproduction with immune response or with genotype-dependent mortality rate. This modelling approach allows us to investigate the competition of different strains and the emergence of resistant strains due to treatment.

In this work we will study the question about the emergence of new strains. From the modelling point of view, these two cases, existence of stable strains and emergence of new strains

are complementary, they are not observed at the same time. The former corresponds to stable pulses while the latter to periodic travelling waves. A nonlocal reaction–diffusion equation describing the emergence of biological species was suggested in [22]. It represents a particular case of Equation (1) without immune response and genotype-dependent mortality. We will show that immune response plays an important role in the dynamics of virus quasi-species.

## 2. Bifurcations of Periodic Structures

An important property of nonlocal reaction–diffusion equations is that the homogeneous in-space stationary solution can lose its stability with respect to spatial perturbations leading to the emergence of periodic solutions. This property was revealed and studied in some models of population dynamics [22,39]. Here we will study it for the models of infection development described by the equation

$$\frac{\partial u}{\partial t} = D\frac{\partial^2 u}{\partial x^2} + ru(1 - qJ(u)) - uf(S(u_\tau)) \tag{1}$$

similar to Equation (1) but without the genotype-dependent mortality term. We will analyze various particular cases of this equation.

### 2.1. Single Nonlocal Term

We consider the reaction–diffusion equation

$$\frac{\partial u}{\partial t} = D\frac{\partial^2 u}{\partial x^2} + ru(1 - qJ(u)) - f(u)u \tag{2}$$

for all real $x$, where

$$J(u) = \int_\infty^\infty \phi(x - y)u(y, t)dy,$$

the function $\phi(x)$ is bounded and non-negative. We will suppose that $\int_{-\infty}^\infty \phi(x)dx = 1$. In what follows, we set $r = q = 1$. Let $u_0 > 0$ be a solution of the equation

$$f(u) = 1 - u. \tag{3}$$

Then $u_0$ is a stationary solution of Equation (2). We will study stability of this stationary solution.

To linearize Equation (2) about $u = u_0$, we look for its solution in the form $u = u_0 + ve^{\lambda t}$, where $v$ is a small perturbation and we obtain the eigenvalue problem

$$Dv'' - u_0(J(v) + f'(u_0)v) = \lambda v. \tag{4}$$

Applying the Fourier transform, we get

$$\lambda = -D\xi^2 - u_0(\tilde{\phi}(\xi) + f'(u_0)), \tag{5}$$

where $\tilde{\phi}(x)$ is the Fourier transform of the function $\phi(x)$. If we replace $\phi(x)$ by the $\delta$-function, instead of (5) we have

$$\lambda = -D\xi^2 - u_0(1 + f'(u_0)). \tag{6}$$

Assuming that

$$1 + f'(u_0) > 0, \tag{7}$$

we conclude from (6) that $\lambda < 0$ for all real $\xi$.

Let us now analyze equality (5). Since $\tilde{\phi}(0) = 1$, then $\lambda(0) < 0$. Suppose that $\phi(x)$ is an even function, $\phi(x) = \phi(-x)$ for all $x$. Then

$$\tilde{\phi}(\xi) = \int_{-\infty}^{\infty} \phi(x)\cos(\xi x)dx,$$

$\tilde{\phi}(0) = 1$, and $\tilde{\phi}(\xi) < 1$ for all $\xi \neq 0$. Assuming that $\lambda = 0$ in (5) for some $\xi$, we obtain the stability boundary:

$$\tilde{\phi}(\xi) = -f'(u_0) - D\xi^2/u_0. \tag{8}$$

This equality should be satisfied for some real $\xi$. Its value is related to the wave number of the corresponding eigenfunction. If we consider a bounded interval with periodic boundary conditions, then $\xi = 2\pi k/L$, where $L$ is the length of the interval and $k = 1, 2, 3 \dots$

### 2.2. Examples

Consider the functions

$$\phi_1(x) = \sqrt{\frac{a}{\pi}}e^{-ax^2}, \quad \phi_2(x) = \frac{a}{2}e^{-a|x|}, \quad \phi_3(x) = \begin{cases} 1/(2N) & , \quad |x| \leq N \\ 0 & , \quad |x| > N \end{cases}.$$

Then

$$\tilde{\phi}_1(\xi) = e^{-\xi^2/(4a)}, \quad \tilde{\phi}_2(\xi) = \frac{a^2}{a^2 + \xi^2}, \quad \tilde{\phi}_3(\xi) = \frac{1}{\xi N}\sin(\xi N).$$

Figure 1 (left) shows a graphical solution of Equation (8) for the function $\phi_3(x)$. The curves corresponding to the functions in the left-hand side and in the right-hand side of this equation touch each other. The corresponding values of parameters belong to the stability boundary. For lesser values of the diffusion coefficient, the homogeneous in-space stationary solution $u_-$ loses its stability resulting in the emergence of a periodic in-space solution.

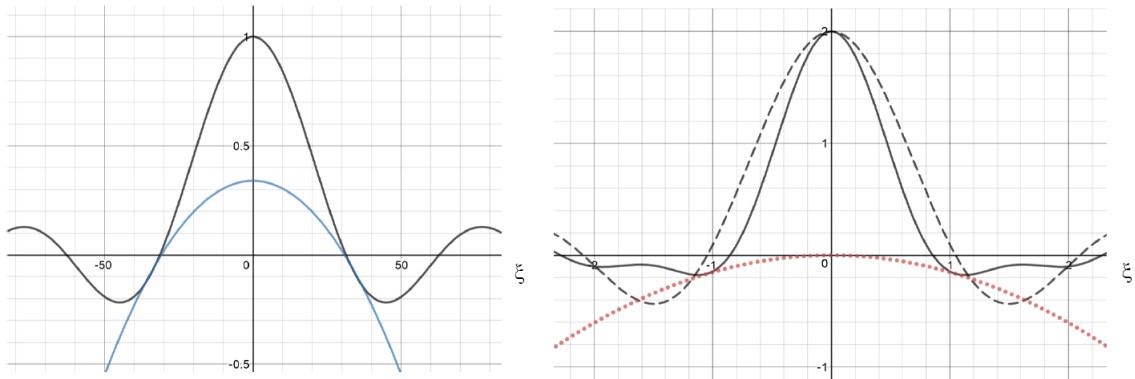

**Figure 1.** (**Left**) Graphical solution of Equation (8): functions $(\sin(\xi N))/(\xi N)$ and $-f'(u_0) - D\xi^2/u_0$, where $N = 0.1$, $f'(u_0) = -0.34$, $D = 0.00033$, $u_0 = 0.93$. (**Right**) Graphical solution of Equation (11): the function $\phi(\xi) + b\tilde{\psi}(\xi)$ for the values of parameters $b = 1$, $N_1 = 3$, $N_2 = 5$ (solid line), $N_1 = 3$, $N_2 = 3$ (dashed line), and the function $-D/u_-\xi^2$ with $D/u_- = 0.1508$ (point line).

Let us note that Fourier transforms of the functions $\phi_1(x)$ and $\phi_2(x)$ are positive. Therefore, if $f'(u_-) \geq 0$, then Equation (8) does not have solution, and solution $u_-$ is stable. If $f'(u_-) < 0$, it has a solution for sufficiently small values of the diffusion coefficient. Thus, emergence of periodic solutions is determined by the interaction of virus mutations, nonlocal competition for host cells and immune response. In terms of virus population distribution in the space of genotypes, these periodic solutions correspond to different virus strains.

### 2.3. Double Nonlocal Equation

Consider Equation (1) with two nonlocal terms $J(u)$ and $S(u)$ and without time delay ($\tau = 0$). For simplicity of presentation, we set $f(u) = bu$. In the stationary case, we obtain the equation

$$Du'' + u(1 - J(u) - bS(u)) = 0.$$

Let us recall that $J(u) = \int_{-\infty}^{\infty} \phi(x-y)u(y)dy$, $S(u) = \int_{-\infty}^{\infty} \psi(x-y)u(y)dy$. Assuming that $\int_{-\infty}^{\infty} \phi(y)dy = \int_{-\infty}^{\infty} \psi(y)dy = 1$, we get a homogeneous in-space stationary solution $u_- = 1/(1+b)$ of this equation. Linearizing the equation about this stationary solution, we obtain the eigenvalue problem:

$$Dv'' - u_-(J(v) + bS(v)) = \lambda v. \tag{9}$$

Applying the Fourier transform to (9), we get

$$\lambda = -D\xi^2 - u_- \left( \tilde{\phi}(\xi) + b\tilde{\psi}(\xi) \right). \tag{10}$$

Consider the functions

$$\phi(x) = \begin{cases} 1/(2N_1) & , \quad |x| \le N_1 \\ 0 & , \quad |x| > N_1 \end{cases}, \quad \psi(x) = \begin{cases} 1/(2N_2) & , \quad |x| \le N_2 \\ 0 & , \quad |x| > N_2 \end{cases}$$

Then

$$\tilde{\phi}(\xi) = \frac{1}{\xi N_1} \sin(\xi N_1), \quad \tilde{\psi}(\xi) = \frac{1}{\xi N_2} \sin(\xi N_2).$$

Stability boundary is determined by the equation

$$\tilde{\phi}(\xi) + b\tilde{\psi}(\xi) = -D\xi^2/u_- \tag{11}$$

obtained from Equation (10) with $\lambda = 0$. An example of graphical solution of this equation is shown in Figure 1. The stability boundary corresponds to the case where the functions in the left-hand side and in the right-hand side of this equation touch each other (solid and point lines). If they intersect (dashed and point lines), then the stationary solution is unstable. For the fixed values of $N_1$, $N_2$ and $b$, stability conditions are determined by the diffusion coefficient.

**Proposition 1.** *For any positive values $N_1$, $N_2$ and $b$ there exists a critical value $D_c$ of the diffusion coefficient such that the stationary solution $u_- = 1/(1+b)$ is stable for $D > D_c$ and unstable for $D < D_c$.*

The proof of this proposition is straightforward. It is sufficient to note that the function $\tilde{\phi}(\xi) + b\tilde{\psi}(\xi)$ has negative values for any values of parameters.

Dependence of the stability conditions on $N_1$ and $N_2$ is more complex. Both can have stabilizing or destabilizing effect on the solution. In the example in Figure 1, decreasing the value $N_2$ leads to the instability of the solution (solid and dashed lines).

*2.4. Delay Equation*

Stability of solutions of the delay equation

$$\frac{\partial u}{\partial t} = D\frac{\partial^2 u}{\partial x^2} + ru(1 - qu) - f(u_\tau)u \tag{12}$$

without nonlocal terms was studied in [37]. Spatial perturbations of the homogeneous in-space solution can lead to a complex spatiotemporal behavior with standing wave, travelling waves and aperiodic dynamics.

Nonlocal delay equation.

Consider now Equation (1) with a single nonlocal term and with time delay:

$$\frac{\partial u}{\partial t} = D\frac{\partial^2 u}{\partial x^2} + ru(1 - qJ(u)) - uf(u_\tau). \tag{13}$$

In what follows, we set $r = q = 1$. Linearizing this equation about a stationary solution $u = u_0$, we obtain the eigenvalue problem

$$Dv'' - u_0\left(J(v) + f'(u_0)e^{-\lambda\tau}v\right) = \lambda v.$$

Applying the Fourier transform, we get

$$\lambda = -D\xi^2 - u_0\left(\tilde{\phi}(\xi) + f'(u_0)e^{-\lambda\tau}\right).$$

For $\lambda = 0$ we obtain Equation (8) for the stability boundary. Therefore, as it can be expected, time delay does not influence the bifurcation of stationary solution. Consider now the bifurcation of time periodic solution. We set $\lambda = i\nu$, where $\nu$ is a real number. Then separating the real and imaginary part in the last equation we obtain:

$$\nu = u_0 f'(u_0)\sin(\nu\tau), \quad D\xi^2 + u_0\tilde{\phi}(\xi) + u_0 f'(u_0)\cos(\nu\tau) = 0.$$

Set $z = \nu\tau$. Then

$$\cos z = -\frac{D\xi^2/u_0 + \tilde{\phi}(\xi)}{f'(u_0)}, \quad \tau = \frac{z}{u_0 f'(u_0)\sin z}. \tag{14}$$

We find the value of $z$ from the first equation and the value of $\tau$ from the second equation. The first equation has a solution if and only if

$$\left|\frac{D\xi^2/u_0 + \tilde{\phi}(\xi)}{f'(u_0)}\right| \leq 1. \tag{15}$$

In the case of the local equation where $\tilde{\phi}(\xi) = 1$, the minimum of the numerator is reached at $\xi = 0$. Therefore, the loss of stability occurs with the space independent perturbations assuming that $|f'(u_0)| < 1$. For the nonlocal equation the loss of stability can occur for $\xi \neq 0$ and for $|f'(u_0)| \geq 1$.

Consider the case where the function $f(u)$ is linear, $f(u) = k_1 u$. If $k_1 > 1$ and $N_1$ is sufficiently small, then the homogeneous in-space solution $u_0$ can lose its stability with respect to temporal perturbation for the time delay $\tau$ large enough. If $\tau$ is less than a critical value then temporal and spatial perturbations decay (Figure 2, left). If we increase $N$ for the other parameters fixed, then the constant solution $u_0$ can lose its stability with respect to spatiotemporal perturbations. Figure 2 (right) shows the emergence of a spatiotemporal pattern at the center of the interval. It propagates and gradually fills the whole spatial domain. Let us now take $k_1 < 1$. Then time oscillations in the local problem ($N_1 = 0$) decay for any time delay. In the nonlocal problem and $N_1$ sufficiently large various spatiotemporal patterns can be observed (Figure A1 in Appendix A).

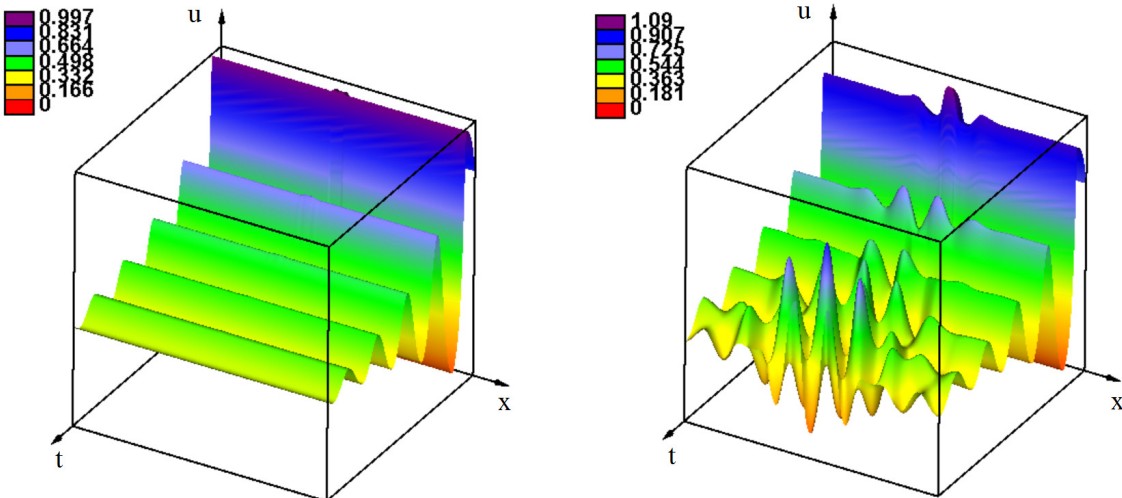

**Figure 2.** Numerical simulations of Equation (13) for the linear function $f(u) = k_1 u$. Spatial and temporal perturbation decay if the solution $u_-$ is stable (**left**). The spatial perturbation at the center of the interval leads to the emergence of a spatiotemporal pattern propagating from the center and gradually filling the whole spatial domain (**right**). The value of parameters: $r = 1, q = 1, k_1 = 1.5, D = 10^{-5}, \tau = 3, N_1 = 0.01$ (**left**) and $N_1 = 0.1$ (**right**), $t = 50$. Here and in all figures below, $L = 1$, unless another value is indicated.

## 3. Emergence of Strains as Periodic Wave Propagation

### 3.1. Propagation Of Waves

We study in this section propagation of described by Equation (1) assuming for simplicity that the second integral term $S(u)$ becomes local, i.e., the kernel $\psi(x)$ is the replaced by the $\delta$-function:

$$\frac{\partial u}{\partial t} = D\frac{\partial^2 u}{\partial x^2} + ru(1 - qJ(u)) - uf(u_\tau). \tag{1}$$

Local and delay equations.

To study the behavior of solutions, we begin with the local case. Then we get conventional reaction–diffusion equation

$$\frac{\partial u}{\partial t} = D\frac{\partial^2 u}{\partial x^2} + F(u), \tag{2}$$

where the function $F(u) = ru(1 - qu) - uf(u)$ can have different numbers of zeros depending on the values of parameters. Besides the zero $u_+ = 0$, there can exist up to three positive zeros, the maximal zero $u_-$ and possibly one or two intermediate zeros $u_1$ and $u_2$.

*Monostable case.* If there is only one positive zero $u_-$, then $F'(u_+) > 0$, $F'(u_-) < 0$. The $[u_+, u_-]$-waves, i.e., the waves with the limits $u(\pm\infty) = u_\pm$ at infinity, exist for all values of the speed greater than or equal to some minimal speed $c_0$. These waves are stable in appropriate weighted spaces [24].

*Bistable case.* In the bistable case, $F'(u_\pm) < 0$, and there is an additional zero $u_1 \in (u_+, u_-)$. The $[u_+, u_-]$-wave exists for a single value of speed $c_1$, and this wave is globally asymptotically stable.

*Monostable–bistable case.* In this case, there are two intermediate zeros, $u_1, u_2, u_1 < u_2$, and $F'(u_+) > 0$, $F'(u_1) < 0$, $F'(u_2) > 0$, $F'(u_-) < 0$. The monostable $[u_+, u_1]$-waves, i.e., the waves with the limits $u(\pm\infty) = u_\pm$ at infinity, exist for all values of the speed greater than or equal to some minimal speed $c_0$. The bistable $[u_1, u_-]$-wave exists for a single value of speed $c_1$. If $c_1 > c_0$, then there exist $[c_+, c_-]$-waves for all speeds $c \in [c_0, c_1)$. If $c_1 \leq c_0$, then such waves do not exist, and there is a system of two waves propagating one after another with different speeds and a growing distance between

them. All these properties including convergence of solutions of the Cauchy problem to waves and systems of waves can be found in [40].

Delay reaction–diffusion equation

$$\frac{\partial u}{\partial t} = D\frac{\partial^2 u}{\partial x^2} + ru(1 - qu) - uf(u_\tau) \tag{3}$$

is a particular case of Equation (1) where the integral $J(u)$ is replaced by $u$. In numerical simulations this equation is considered in a sufficiently long interval $0 < x < L$ with the homogeneous Neumann boundary conditions and with some initial condition $u(x,t) = u_0(x), -\tau \leq t \leq 0$, where $u_0(x) = u_0$ for $0 \leq x \leq x_0$ and $u_0(x) = 0$ otherwise. Here $u_0$ and $x_0$ are some positive constants. Equation (3) was introduced in [27] to study spatial models of infection development in tissues. The influence of time delay on the wave propagation manifests itself in the most spectacular way in the monostable–bistable case where $c_0 > c_1$, and there is a system of two waves propagating with different speeds. The presence of time delay can lead to the emergence of complex spatiotemporal structures between the two waves. Some examples of numerical simulations are shown in Figure 3.

Existence of waves described by this equation was proved in [41,42]. Since conventional monotonicity conditions and the maximum principle are not applicable in this case, the proof of the wave existence requires sophisticated mathematical techniques. There are only a few works where the wave existence is proved for the delay reaction–diffusion equation in the bistable case without the monotonicity condition (see also [33]).

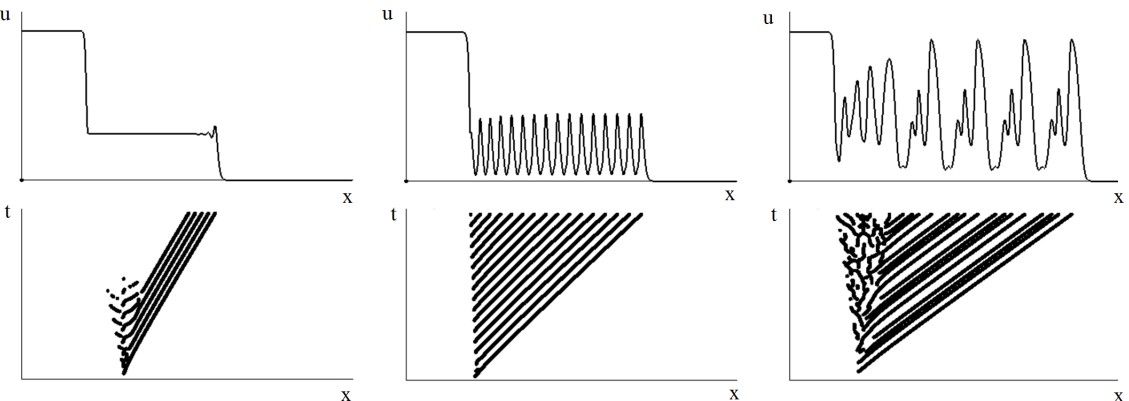

**Figure 3.** Snapshots of different regimes of wave propagation in numerical simulations of Equation (3) in the monostable–bistable case. The speed of the monostable wave is greater than the speed of the bistable wave, and the distance between them grows (**upper row, left**). The intermediate equilibrium between the wave becomes unstable, and the monostable wave is space periodic (**upper row, middle**). This periodic wave can be followed by complex spatiotemporal oscillations (**upper row, right**). The lower row shows the position of local maxima of the same solutions on the $(x,t)$-plane. Reprinted from [37] with permission.

### 3.1.1. Nonlocal Equation

The presence of the nonlocal term in Equation (1) can influence the regimes of wave propagation presented above for the local equation. Let us recall that the homogeneous in-space stationary solution $u_-$ can be stable or unstable depending on the values of parameters. In particular, for a sufficiently small diffusion coefficient or for a sufficiently large $N$ in the definition of the kernel $\phi(x)$:

$$\phi(x) = \frac{1}{2N}\begin{cases} 1 &, \quad |x| \leq N \\ 0 &, \quad |x| > N \end{cases} \tag{4}$$

this solution loses its stability resulting in the bifurcation of a periodic in-space stationary solution. If this solution is stable, then the regimes of wave propagation are the same as before (monostable, bistable, monostable–bistable). Suppose now that it is unstable and consider, first, the monostable case. Then there are two transition: the first one is provided by the $[u_+, u_-]$-wave, the second one is the transition from the constant solution $u_-$ to the periodic solution $u_p(x)$. If the speed $c_0$ of the former is greater than the speed $c_p$ of the latter, then they propagate one after another one with a growing distance between them (Figure 4, left). If $c_p > c_0$, then they merge and form a single periodic wave (Figure 4, right). These different regimes are also observed in the bistable and monostable–bistable cases.

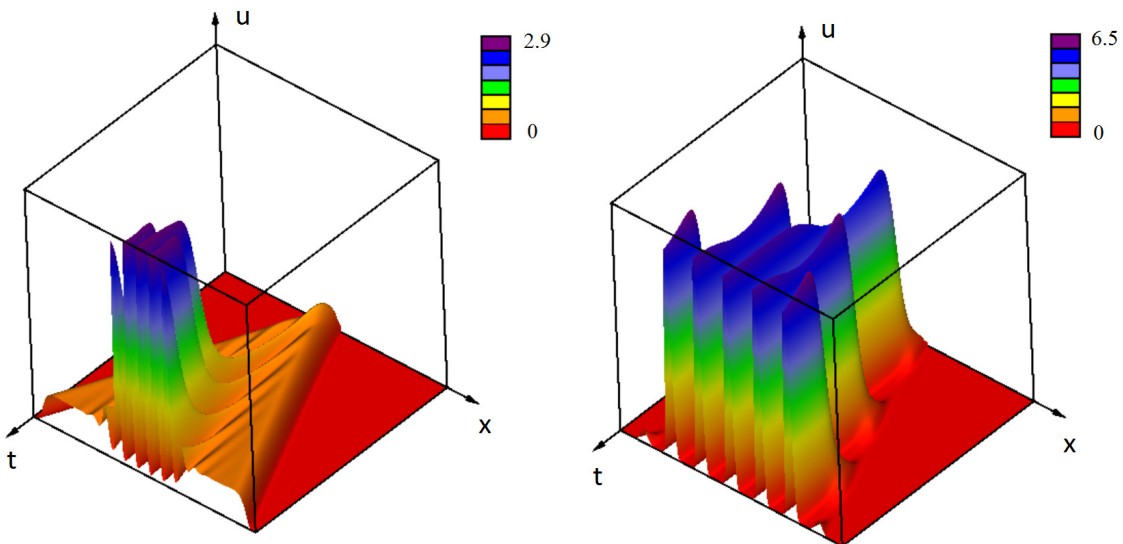

**Figure 4.** Numerical simulations of Equation (1) show the waves propagating from the center of the interval towards its boundaries in the monostable case. In the first monostable case (**left**) the periodic perturbation propagates slower than the $[u_+, u_-]$-wave, and the distance between them grows. In the second monostable case (**right**), the periodic perturbation propagates faster, it merges with the wave, and they form a single periodic wave. The values of parameters: $D = 10^{-5}, r = 1, q = 1, k_1 = 5, k_3 = 3$, $N = 0.035$ (**left**), $N = 0.1$ (**right**); $f(u) = k_1 u e^{-k_3 u}, \tau = 0$.

Next, consider Equation (1) with time delay and, for simplicity of presentation, with a linear function $f(u)$. In this case, we have a monostable equation with two stationary points $u_+ = 0$ and $u_- > 0$. Stability analysis of the homogeneous in-space stationary solution $u_-$ with respect to spatial and temporal perturbations was carried out in Section 2. If this point is stable, then we observe propagation of a usual $[u_+, u_-]$-wave with a constant speed and a constant profile. However, this wave is not necessarily monotonic with respect to $x$, as it is the case for the local equation. Damped oscillation behind the wave occur for $N$ sufficiently large (Figure 5, left).

Behavior of solutions becomes more complex if the solution $u_-$ is unstable. If the spatiotemporal perturbation of this solution propagates in space with the speed less than the speed of the $[u_+, u_-]$-wave, then this wave propagates with a constant speed and a constant profile, possibly with decaying spatial oscillations behind the wave front. This wave is followed by the region of spatiotemporal oscillations (Figure 5, right). If the perturbations of the solution $u_-$ propagate faster than the $[u_+, u_-]$-wave, then they merge and form an oscillating wave propagating with a variable speed (not shown).

Let us recall that in the monostable case the waves exist for all values of the speed greater than or equal to the minimal speed $c_0$. The value of the minimal speed is determined by the linearized problem at 0, and it does not depend on $N$ and $\tau$ if the wave remains stable.

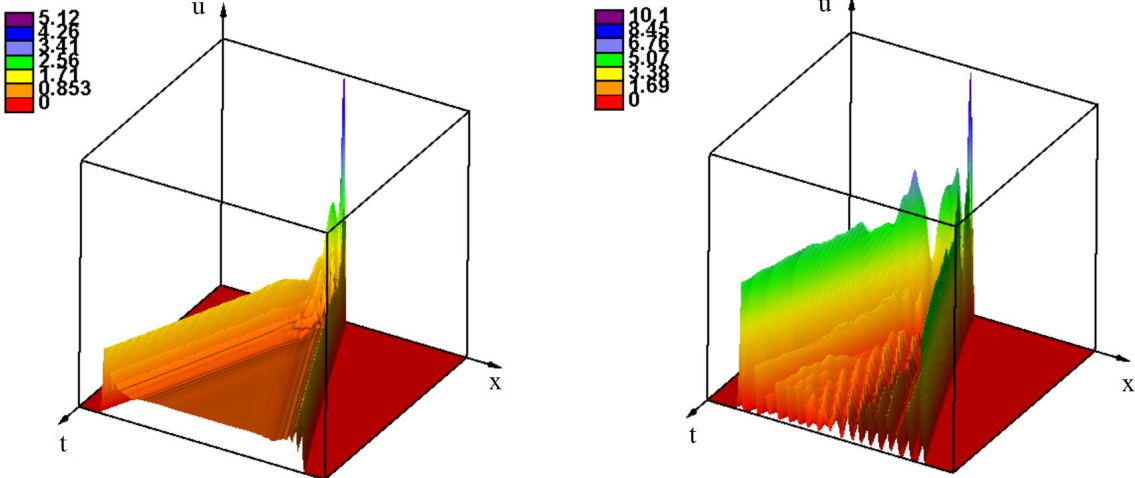

**Figure 5.** Numerical simulations of Equation (1) with $f(u) = k_1 u$. If the solution $u_-$ is stable, then there is a $[w_+, w_-]$-wave propagating with a constant speed and profile with possible spatial oscillations independent of time (**left**). If this solution is unstable, then this wave is followed by spatiotemporal oscillations (**right**). The values of parameters: $r = 1, q = 1, L = 2, D = 10^{-5}, k_1 = 0.9, \tau = 3, N = 0.25$, $\tau = 2$ (**left**), $\tau = 4$ (**right**), $t = 150$.

### 3.1.2. Bifurcations of Waves and Pulses

Existence and stability of pulses and waves for Equation (1) depends on the width $N$ of the support of the kernel $\phi(x)$.

Let us recall that the local reaction–diffusion equation with a bistable function $F(u)$ has a positive stationary solution decaying at infinity (pulse solution) if and only if $I_F = \int_{u_+}^{u_-} F(u)du > 0$. This stationary solution is unstable. It also has a stable $[u_+, u_-]$-wave whose speed is positive under the same condition on the integral $I_F$. Similar properties hold for the nonlocal equation with $N$ sufficiently small. With an increase of $N$, the wave becomes non-monotone as a function of $x$ but it still has a constant speed and profile.

Periodic structures and waves appear as the width $N$ of the support of the kernel $\phi(x)$ exceeds a critical value $N_c^1$ for which the corresponding eigenvalue of the linearized problem crosses the origin. If we further increase the value of $N$, then instead of periodic waves we observe stable pulses. Thus, there are two bifurcations with a transition from simple waves to periodic waves (through an intermediate regime with two waves) and from periodic waves to pulses. The first bifurcation occurs due to the essential spectrum crossing the origin. The second one is a nonlocal bifurcation where the speed of the periodic wave decreases as $N$ approaches a critical value $N_c^2$, and it becomes zero for $N$ exceeding the critical value. At the same time, the spikes of the periodic wave become pulses. Let us note that multiple pulses are not stationary solutions of Equation (1), they slowly move from each other with a decaying speed.

### 3.2. Emergence of Strains

Virus density distribution $u(x, t)$ as a function of its genotype $x$ and time $t$ characterizes the existence of virus strains and their evolution. In this context, a strain is a positive localized solution of Equation (1), i.e., a solution with maximum at some $x_0$ (most frequent genotype) and rapidly decaying as the distance $|x - x_0|$ increases. Existence and stability of stationary localized solutions (pulses) of Equation (1) was studied in [23]. They correspond to persisting virus strains. In this section, we will study the emergence of new strains due to propagating of periodic waves. As it was discussed in the previous section, stable pulses and waves are mutually exclusive.

### 3.2.1. Initiation of Periodic Waves

Suppose that the initial virus distribution $u_0(x)$ represents a non-negative function with a narrow support at the center of the interval. For a properly chosen values of parameters the solution of Equation (1) with this initial condition develops to a periodic travelling wave. At the first stage of this dynamics, the solution rapidly growth remaining localized at the center of the interval (Figure 6a). It reaches a maximal level, and then the peak gradually decreases and becomes wider (Figure 6b,c). At some moment of time, two other peaks appear from each side of the first one. After a short transient period, they converge to approximately the same height. If the interval is sufficiently large, then other peaks will appear after some time gradually filling the whole interval (Figure 4, right). This is a typical dynamics of the initiation of periodic waves which occurs under the conditions presented in Section 2.

Let us also note that new strains (peaks) appear at some distance from the first one by a genetic jump and not as a gradual evolution of the original strain. The value of the virus density between them is close to zero. This is not the case for a large diffusion coefficient (mutation rate) where new strains appear continuously (Appendix A, Figure A2). If the diffusion coefficient is large enough, then the strains do not form, and viruses with any genotype exist. This is determined by the stability of the stationary points (Section 2).

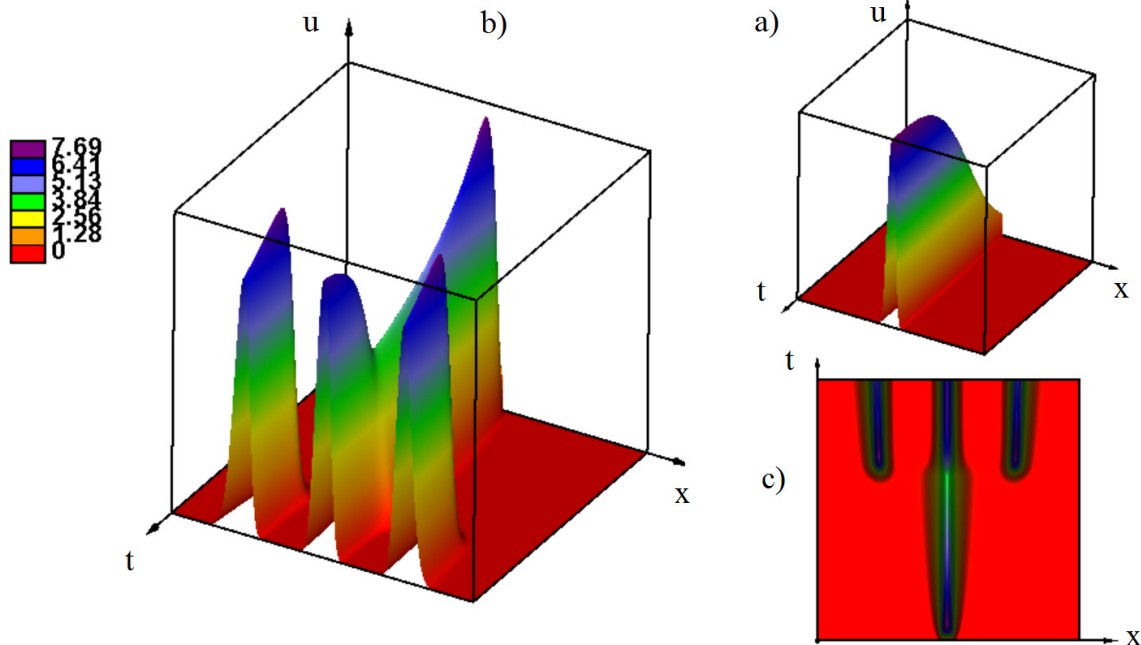

**Figure 6.** Emergence of a periodic wave in numerical simulations of Equation (1). (**a**) At the first stage, solution growth remaining localized at the center of the interval. (**b**) Then it decreases and widens, and after some time, other peaks of solution appear. (**c**) Another representation of the same solution as in (**b**). Values of parameters: $D = 10^{-5}, r = 1, q = 1, N = 0.2, \tau = 0, f(u) = 0$, the maximum of the initial condition 0.9, $t = 75$.

### 3.2.2. the Influence of Immune Response

We consider the function of immune response in the form $f(u) = (k_1 u + k_2)e^{-k_3 u}$. In order to explain the influence of immune response on the emergence of strains, consider the function $F(u) = ru(1 - qu - f(u))$. If $k_2 = k_3 = 0$, then $F(u)$ has a single positive zero $u_-$, and this is a monostable case. For the values of $k_1$ sufficiently small, behavior of solution of Equation (1) is similar to the case where $f(u) \equiv 0$ with the propagation of a periodic wave and the emergence of new strains (Figure 6). If $k_1$ is large enough, then the equilibrium $u_-$ becomes stable, and there is a stationary $[u_+, u_-]$-wave without emerging peaks, as it is the case of a periodic wave.

Let us recall that the growing branch of the function $f(u)$ corresponds to the antigen stimulated immune response while the decreasing characterizes death or exhaustion of immune cells due to high virus concentration. Thus, if we consider only the growing branch, then a strong immune response (large $k_1$) does not eliminate infection but prevents the formation of virus strains. Instead of the localized solutions with separated peaks, the virus density distribution converges to a constant positive solution.

In the case where the decreasing branch of the immune response function is present ($k_2 = 0$, $k_3 \neq 0$), the function $F(u)$ can have up to three positive zeros $u_1 < u_2 < u_-$. As we discussed above, this is a monostable–bistable case where the behavior of solutions depends on the values of parameters and on the choice of initial condition. Set $u_0(x) = u_0$ for $x_1 \leq x \leq x_2$ and $u_0(x) = 0$ otherwise. If $u_0$ is small enough, then the solution represents a monostable wave (Figure 7, left) without strain formation. If $u_0$ is sufficiently large, then for the same values of parameters as before, the central peak is formed followed after some time by the appearance of two monostable waves of low amplitude (Figure 7, right).

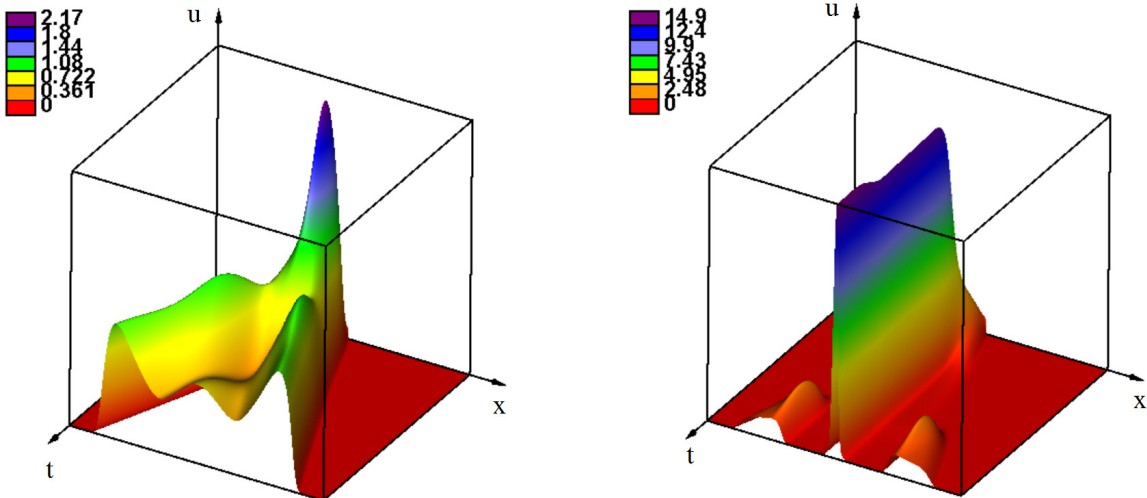

**Figure 7.** Numerical simulations of Equation (1) with two different initial conditions and the same values of parameters: $D = 10^{-5}, r = 1, q = 1, N = 0.2, \tau = 0, f(u) = k_1 e^{-k_3 u}$, $k_1 = 1, k_3 = 0.6$, the maximum of the initial condition equals 0.1 (**left**) and 0.9 (**right**), $x_1 = 0.48, x_2 = 0.52, t = 75$.

Furthermore, the width $x_2 - x_1$ of the support of the initial condition can also influence this behavior. A counterintuitive result is that increasing the support of the initial condition leads to the disappearance of the high amplitude peak and to the convergence of solution to the low amplitude monostable wave. The explanation of this effect is that two pulses (peaks) form if the support is sufficiently wide. They compete with each other, their amplitude becomes less than for a single pulse, it is not sufficient to overcome the threshold and to form a stable central pulse.

Under further increase of $k_3$, propagation of a periodic wave, as it is described above, is observed. If $k_2 \neq 0$, then $f(0) > 0$, i.e., immune response is nonzero even without antigen due to memory cells. Depending on the values of parameters, solution can form a stable pulse, vanish, or initiate simple or periodic waves described above.

### 3.2.3. Effect of the Delay of the Antiviral Immune Response

In the case of a nonlocal equation with time delay in the immune response term, spatial structures presented in the previous paragraph can become oscillating. Some examples of virus strain evolution are shown in Figure 8. The left and middle figures are obtained for the same values of parameters with different initial conditions. If the initial virus load is large enough, then there is a dominating virus strain and some other strains with low virus density and a complex spatiotemporal behavior.

If the initial virus load is sufficiently small, then the dominating central strain does not exist, and there is only a variety of different genotypes with low densities. Changing the properties of the immune response (function $f(u)$), we observe stable stationary strains similar to those in Figure 4 (right image) after the initial front propagation.

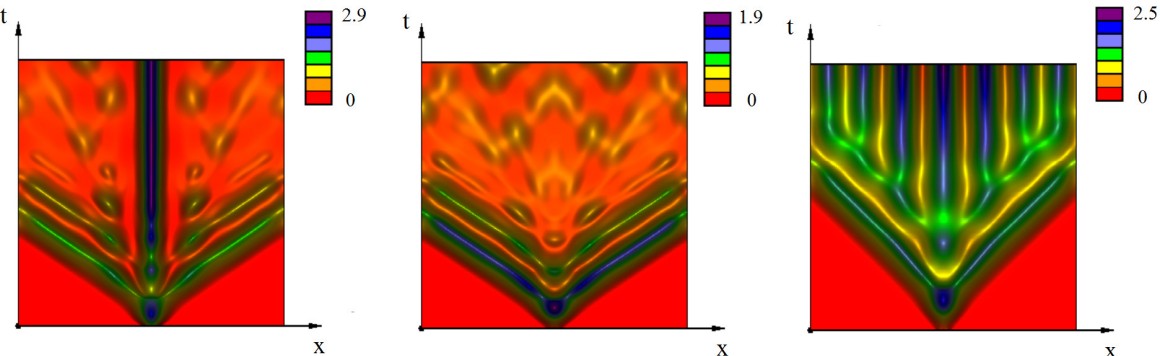

**Figure 8.** Numerical simulations of Equation (1). Virus evolution with time delay in the term describing the immune response represented as level lines of the solution $u(x, t)$ on the $(x, t)$-plane. Different regimes coexist for the same values of parameters depending on the initial conditions, with high initial viral load (**left**) and low initial viral load (**middle**). Values of parameters: $D = 10^{-4}, r = 1, q = 1$, $N = 0.1, f(u) = k_1 e^{-k_3 u}, k_1 = 8, k_3 = 3, t = 80$ (**left** and **middle**), $k_3 = 6, t = 50$ (**right**); the maximum of the initial condition 0.9 (**left**), 0.1 (**middle** and **right**).

### 3.2.4. the Influence of Genotype-Dependent Mortality

We will finish this section with the analysis of the genotype-dependent mortality on the emergence and evolution of virus quasi-species. In order to show this influence more precisely, we consider it in the case without immune response, $f(u) \equiv 0$. Set $\sigma(x) = 0$ for $x_1^* \leq x \leq x_2^*$ and $\sigma(x) = 1$ otherwise. Figure 9 (left) shows the emergence of virus strains for $x_1^* = 0.3, x_2^* = 0.7$ and $N = 0.09$. The initial condition has a support at the center of the interval. Similar to the case of initiation of a periodic wave, there is one strain in the beginning of the simulation, and two other strain appear sometime later. These three strains fill the whole admissible interval where $\sigma(x) = 0$, and new strains do not appear outside of this interval because virus mortality rate is greater there that its reproduction rate.

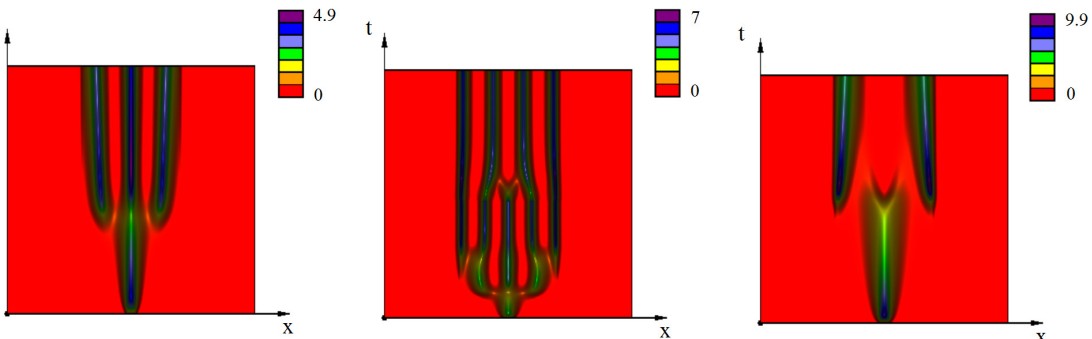

**Figure 9.** Numerical simulations of Equation (1). Virus evolution without immune response and with the genotype-dependent mortality $\sigma(x)$ represented as level lines of the solution $u(x, t)$ on the $(x, t)$-plane. Values of parameters: $D = 10^{-5}, r = 1, q = 1, N = 0.09$ (**left**), $N = 0.08$ and 0.09 (**middle**), $N = 0.2$ (**right**).

In the middle figure, we begin the simulation with $N = 0.08$ with five emerging strains. When they become steady and do not evolve any more, we change the value of $N$ to $N = 0.09$, as in the previous simulation. However, this time we observe the regime with four strains instead of three strains

observed previously. Hence, different stationary regimes can exist for the same values of parameters, and the initial condition determines the convergence to each of them.

Finally, consider the case where $N = 0.2$ (Figure 9, right). As before, there is single peak of solution at the center of the interval in the beginning of the simulation. However, after some time, it disappears giving rise to two other peaks. Such behavior is determined by the size of the admissible interval: it cannot support three wide peaks, and the two of them from the sides suppress the one at the center of the interval. Thus, virus tends to fill the admissible interval in the most efficient way, that is, to maximize its total density.

## 4. Discussion

### 4.1. Virus Quasi-Species

Speciation is considered to be a general property of the living matter [43]. It manifests itself in the emergence of biological species and in a variety of other systems [44]. In the framework of mathematical modelling, speciation appears due to a non-homogeneous density distributions $u(x, t)$. In biological populations, $x$ can be a morphological characteristic or some characterization of the genotype.

Describing virus quasi-species dynamics, we observe some similarities with the general speciation theory due to the competition for host cells but also some specific features because of the presence of the immune response and of the genotype-dependent mortality. If we consider the virus density distribution $u(x, t)$ as a function of its genotype $x$ and of time $t$, then a virus quasi-species (strain or variant genome) corresponds to a localized solution with a maximal value at some genotype $x_0$ and rapidly decaying as the distance $|x - x_0|$ increases. In the case of persistent strains, the most frequent genotype $x_0$ is fixed but it can be also time-dependent for the evolving strains.

Existence of virus strains considered to be a positive stationary density distribution decaying at infinity (in the approximation of an infinite genotype space) is not a priori given. In the previous work [23] we revealed two mechanisms providing the existence and stability of such solutions. In the first case, the existence of virus strains is determined by the genotype-dependent mortality where the virus can survive only inside some viability interval. The maximum of the density distribution is achieved in the middle of the corresponding viability interval. The second mechanism is determined by the immune response under the assumption that the immune response function $f(u)$ decreases for large $u$. In this case, the virus can survive and form a persistent strain if its concentration is sufficiently high, and if the competition with other strains for host cells occurs in a sufficiently wide range of genotypes.

In mathematical terms, virus strains correspond to stable pulse solutions of the corresponding nonlocal reaction–diffusion equations. Existence of stable pulses does not occur for the conventional local equations.

### 4.2. Emergence of New Quasi-Species: Summary of the Results

It is important to note that stable pulses and periodic travelling waves are mutually exclusive, they are not observed for the same values of parameters. In this work we study periodic travelling waves. Emergence of new peaks in the virus density distribution during the wave propagation corresponds to the emergence of new virus strains.

From the mathematical point of view, the conditions of the emergence of periodic travelling waves can be determined by the linear stability analysis of the homogeneous in-space stationary solutions. In order to show the influence of different factors on the stability conditions, this analysis is carried out in Section 2 for the single nonlocal term, for both nonlocal terms and for the nonlocal delay equation. In the presence of a single nonlocal term, periodic spatial structures bifurcate from the constant solution if the diffusion coefficient $D$ is sufficiently small and if the kernel $\phi(x)$ of the

integral satisfies certain conditions. In particular, for a piece-wise constant kernel, its support should be sufficiently large.

In the case of two nonlocal terms, the qualitative behavior of solutions is similar. The interaction of the nonlocal terms can have stabilizing or destabilizing effect depending on the values of parameters. The influence of time delay (and a single nonlocal term) on the stability conditions depends on the immune response function resulting in temporal or spatiotemporal oscillations.

The transition between a virus-free equilibrium and an infected equilibrium is provided by travelling waves (Section 3). In the case without nonlocal terms or time delay, this is a conventional wave with a constant speed and profile or two waves propagating one after another with different speeds. The nonlocal terms can result in the emergence of periodic waves, while time delay can lead to a complex spatiotemporal pattern formation.

Let us note that the qualitative behavior of solutions is quite robust, and it is not very sensitive to the particular choice of the immune response function and of the integral kernels. The range of genotypes is supposed to be sufficiently large to neglect the influence of the boundaries. Mathematically, we can consider the whole real axis. In numerical simulations, we consider a bounded sufficiently large interval. In most cases, we stop the simulations before the wave approaches the boundary of the interval. In some cases (Figure 8), we continue the simulations to reveal spatiotemporal pattern formation inside the interval. In this case, periodic boundary conditions are more convenient because they do not influence the behavior of the solution inside the interval. Dirichlet or Neumann boundary conditions can influence the behavior of solutions. Thus, periodic boundary conditions do not have here biological significance, but they are more appropriate for mathematical modelling.

### 4.3. Biological Interpretations

Suppose that the initial viral load is localized in a narrow interval of genotypes (e.g., a founder virus in the HIV case). Due to its multiplication and mutations, the density distribution grows and widens. Viruses with different genotype begin to compete for the host cells leading to the appearance of new strains at some distance to avoid the competition with the existent strains. This description of speciation is generic [22], it is not specific for virus quasi-species. Specific features of virus diversification and strain emergence are related to the immune response.

If we take into account the cross-reactivity in the immune response, i.e., different antigens can stimulate the same immune cells, then the immune response interferes with virus competition for the host cells. This interaction is quite complex and can act in different ways on different strains. However, if the mutation rate (diffusion in the genotype space) is sufficiently small, then the speciation of virus quasi-species will necessarily occur. The critical conditions leading to the emergence of new strains depend on the parameters of the problem, and the immune response can have both stabilizing and destabilizing effect.

The influence of immune response becomes easier to determine if we neglect cross-reactivity. Assuming that the immune response function $f(u)$ is increasing due to the stimulation of immune response by the virus antigens, the model predicts that immune response acts to suppress the formation of new strains. If the growth rate of the function $f(u)$ is sufficiently high, then the speciation of the virus density distribution completely disappears. In this case, instead of a discrete set of virus strains our study predicts that a uniform density distribution as a function of virus genotype will take place.

The situation becomes again more complex if we consider also a decreasing branch of the immune response function, which appears due virus-induced death of immune cells, and time delay in the immune response required for the clonal expansion of immune cells. In this case, along with periodic travelling waves described above, complex nonlinear dynamics of solutions can take place with various patterns of emerging and disappearing strains.

Genotype-dependent virus mortality restrains the evolution of virus species to the admissible interval. The emergence and the evolution of virus strains within the viability interval depend on the

values of parameters and on the initial viral load. Let us note that different virus density distributions can be observed for the same values of parameters.

Concluding this discussion, we point out the limitations of the model considered in this work due to various simplifying assumptions. We do not take into account the presence of different immune cells and of cytokines participating in the immune response, complex intracellular regulation of cell fate and of virus multiplication. On the other hand, these and other simplifications allow us to reveal some generic properties of the evolution of virus quasi-species, which would be more difficult to identify in a more complex model. This modelling framework provides a starting basis for further investigations and for the introduction of more detailed models.

**Author Contributions:** Conceptualization, G.B. and V.V.; methodology, G.B. and A.M.; software, N.B. and V.V.; formal analysis, V.P. and V.V.; writing—original draft preparation, G.B. and V.V. All authors have read and agreed to the published version of the manuscript.

**Funding:** This research received no external funding.

**Acknowledgments:** The research was funded by the Russian Science Foundation (Grant no. 18-11-00171) to N.B., G.B. A.M. and V.V. V.P. and V.V. were partially supported by the "RUDN University Program 5-100". A.M. was also supported by a grant from the Spanish Ministry of Economy, Industry and Competitiveness and FEDER grant no. SAF2016-75505-R (AEI/MINEICO/FEDER, UE) and the "Maria de Maeztu" Programme for Units of Excellence in R&D (MDM-2014-0370).

**Conflicts of Interest:** The authors declare no conflict of interest.

## Appendix A. Additional Simulations

Patterns bifurcating due to the instability of the homogeneous in-space solutions.

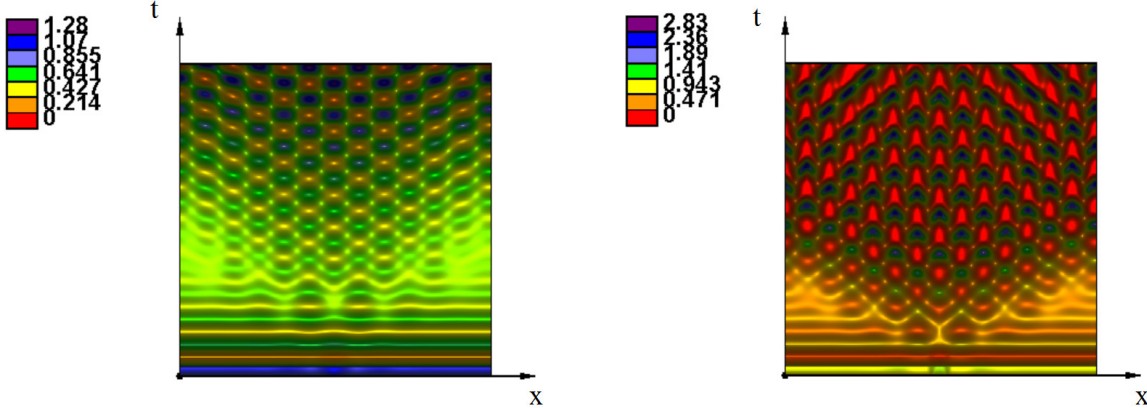

**Figure A1.** Numerical simulations of Equation (13) for the linear function $f(u) = k_1 u$. Level lines of the solution $u(x, t)$ on the plane $(x, t)$ (**left**). Two snapshots of solution (**right**). The value of parameters: $r = 1, q = 1, k_1 = 0.95, N = 0.1, \tau = 4, D = 0.0001$ (**left**), $D = 0.00001$ (**right**), $t = 150$.

Initiation and propagation of periodic waves.

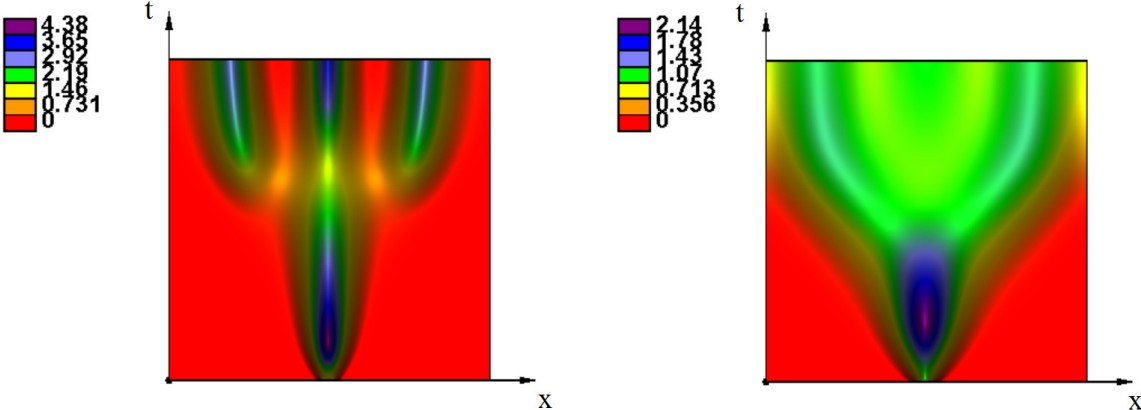

**Figure A2.** Level lines of the solution $u(x,t)$ of Equation (1) on the $(x,t)$-plane. Values of parameters: $r = 1, q = 1, N = 0.2, \tau = 0, f(u) = 0$ (left and middle), the maximum of the initial condition 0.9, $D = 0.0001, t = 35$ (**left**) and $D = 0.0005, t = 20$ (**right**).

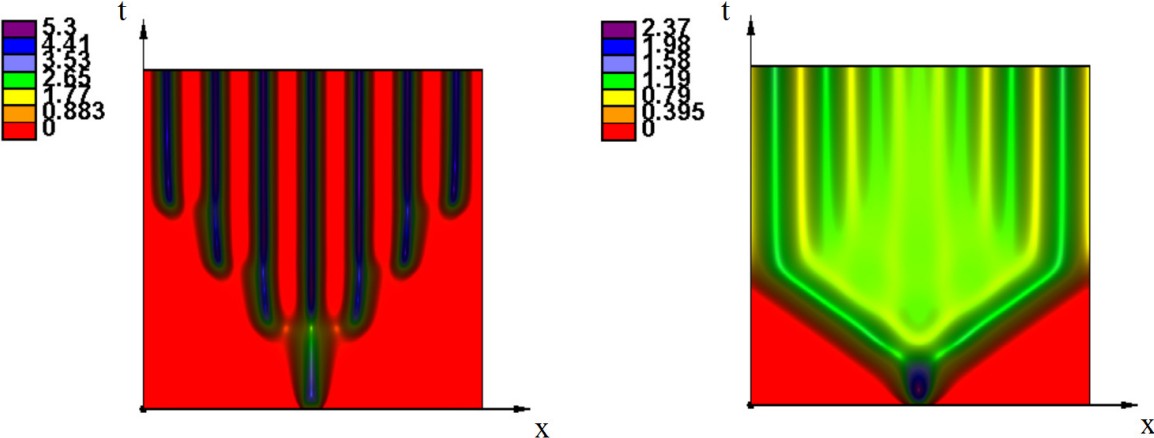

**Figure A3.** Level lines of the solution $u(x,t)$ of Equation (1) on the $(x,t)$-plane. Values of parameters: $r = 1, q = 1, N = 0.1, \tau = 0, f(u) = 0$ (left and middle), the maximum of the initial condition 0.9, $D = 0.00001, t = 130$ (**left**) and $D = 0.0001, t = 75$ (**right**).

Propagation of waves in the case of time delay in the immune response term.

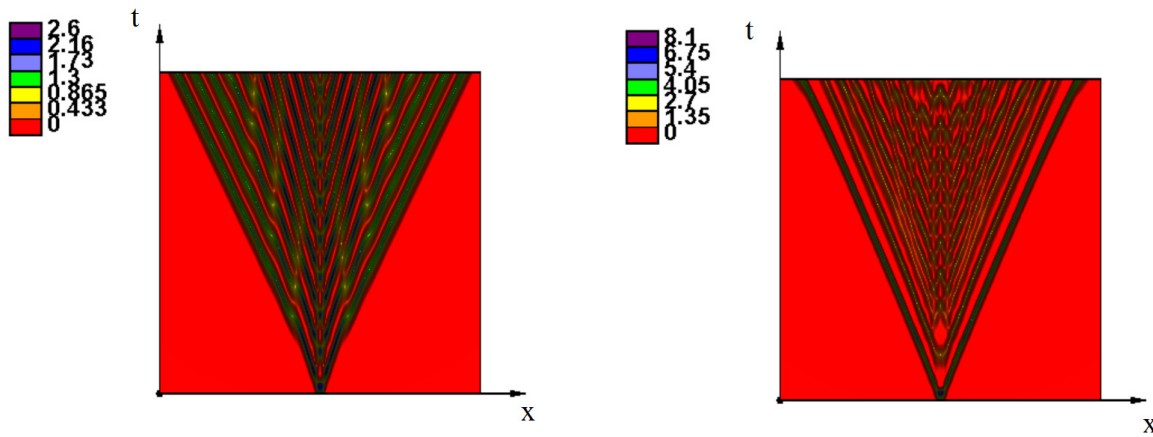

**Figure A4.** Level lines of the solution $u(x,t)$ of Equation (1). The values of parameters: $r = 1, q = 1$, $L = 2, D = 10^{-5}, k_1 = 0.9, \tau = 4, N = 0.05$ (**left**), $N = 0.1$ (**right**), $t = 150$.

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
