# Peer review of "Nonlocal Reaction–Diffusion Model of Viral Evolution: Emergence of Virus Strains"

_mathematics, doi:10.3390/math8010117_

Round 1

Reviewer 1 Report

The paper introduces a mathematical model to study the dynamics of virus evolution. The model is based on a non-local reaction-diffusion differential equation, whose stationary solutions represent virus strains. By performing the stability analysis for such solutions, one can understand the conditions necessary for a stable virus strain to exist, as well as when a new virus strain can emerge. The equation contains two non-local terms, one for the cell reproduction and the other for the immune response. The non-local features of these two terms allow one to examine a more realistic situation that involves the infection capabilities of different viral genotypes.

The study in the paper also improves the current understanding of other related problems. The model is developed upon a well-established virus quasi-species theory by introducing non-local factors in the virus reproduction rate and the immune cell response. This enhancement leads to a more complex model that merits study. In particular, the paper focuses on the cases when new virus strains emerge, providing a detailed analysis and discussions on different scenarios.

My concerns are about the organization of the paper. The main result of the paper is the analysis of the stationary solutions, for the non-local reaction-diffusion differential equation, in different scenarios. However, the case distinctions and their corresponding conclusions are somewhat unclear. For the paper to get its core message across more clearly, I suggest a summary section or a theorem, that explains clearly the core result, without going too deep into discussions about technique.

Minors:

1. There are two equations (1) in the paper, one in line 43 and the other in line 114. One contains a term $\sigma(x)u$ that is not used, or explained in the paper. Please stick with a single one to avoid confusion.

2. The paper has both the term ``quasispecies`` and ``quasi-species``. Please stick with one for consistency.

With my comments addressed, I recommend the paper be accepted for publishing in the Mathematics journal.

Author Response

The paper introduces a mathematical model to study the dynamics of virus evolution. The model is based on a non-local reaction-diffusion differential equation, whose stationary solutions represent virus strains. By performing the stability analysis for such solutions, one can understand the conditions necessary for a stable virus strain to exist, as well as when a new virus strain can emerge. The equation contains two non-local terms, one for the cell reproduction and the other for the immune response. The non-local features of these two terms allow one to examine a more realistic situation that involves the infection capabilities of different viral genotypes.

The study in the paper also improves the current understanding of other related problems. The model is developed upon a well-established virus quasi-species theory by introducing non-local factors in the virus reproduction rate and the immune cell response. This enhancement leads to a more complex model that merits study. In particular, the paper focuses on the cases when new virus strains emerge, providing a detailed analysis and discussions on different scenarios.

My concerns are about the organization of the paper. The main result of the paper is the analysis of the stationary solutions, for the non-local reaction-diffusion differential equation, in different scenarios. However, the case distinctions and their corresponding conclusions are somewhat unclear. For the paper to get its core message across more clearly, I suggest a summary section or a theorem, that explains clearly the core result, without going too deep into discussions about technique.

We have added a summary of the results in the section “Emergence of new quasi-species: summary of the results” in the Discussion.

Minors:

There are two equations (1) in the paper, one in line 43 and the other in line 114. One contains a term $\sigma(x)u$ that is not used, or explained in the paper. Please stick with a single one to avoid confusion.

In the original submission, these two equations had different numbers, (1.1) and (2.1). Section numbers were removed due to formatting by the editorial team. This technical question will be solved with the editorial team.

The last term of equation (1.1) is explained in the paragraph after the equation: “its elimination by immune response and by genotype dependent mortality, either natural or caused by an antiviral treatment.” To make it clearer, we have added the following paragraph in the introduction:

The last term in the right-hand side of equation (1.1) describes virus mortality with the rate depending on its genotype. The viability interval, that is, the rate of genotypes where virus multiplication rate exceeds its mortality can depend on its intrinsic features and on an antiviral treatment.

Stability analysis in Section 2 is carried out without this term. This comment is added after equation (2.1).

Numerical simulations taking into account this term are presented at the end of Section 3.

The paper has both the term ``quasispecies`` and ``quasi-species``. Please stick with one for consistency.

We have changed it everywhere to “quasi-species”.

With my comments addressed, I recommend the paper be accepted for publishing in the Mathematics journal.

We are grateful to the referee for this recommendation.

Reviewer 2 Report

See attachment.

Author Response

This manuscript describes a spatiotemporal model of viral genotype development that includes non-local host cell and immune response interactions. The model is analyzed using both analytical and computational approaches. While the model is novel and examines an important phenomenon, some parts of the manuscript do not clearly connect the math to biology. I therefore recommend that the manuscript undergo major revision.

We are grateful to the referee for this recommendation.

Specific comments:

The manuscript considers a number of possible mathematical formulations for host cell-virus interaction, the immune response, and genotype mortality. There is a lack of biological justification for the mathematical form of these functions. What processes do these functions represent and why look at different functions?

The immune response function f(u) is described in the introduction. Its description is completed in the revised version of the paper. The kernels of the integrals (functions phi and psi) can have three principally different forms: Dirac delta-function, constant function, a positive decreasing function of the modulus of its argument. In the latter case, its exact form in the applications is not known, and different examples will be considered below. Let us note that the qualitative behavior of solutions is quite robust, and it is not very sensitive to the particular choice of the immune response function and of the integral kernels. These comments are added in the introduction and in the discussion.

The section describing mathematical stability analysis of the model (section 2) is not very clear. For example, equation 2.4 suddenly introduces the variable v without explanation of how it is related to the original variable u.

v is a small perturbation of the solution u0. The explanation is added in the text.

It is not clear how the authors arrived at the stationary solution given at the top of page 7. There seems to have been some assumption about the form of functions J(u) and S(u), but these assumptions are not stated in the manuscript.

An explanation is added in the revised version of the paper.

The manuscript is also not very clear on what are new results and previous work. Some specific forms of this model appear to have been analyzed in previous work, so the authors need to be clearer about what is new here (different assumptions for the mathematical form of certain interactions?).

The novelty of this work consists in the investigation of the emergence of new strains described by equation (1.1). From the mathematical point of view, this implies the investigation of periodic travelling waves. This question was not studied before for this model. The explanation is given in the last paragraph of the introduction and in the discussion.

In the last sentence of section 3, what do the authors mean by efficient? 

We mean here that virus tends to maximize its total density. The explanation is added.

Reviewer 3 Report

This manuscript describes a spatiotemporal model of viral genotype development that includes non-local host cell and immune response interactions. The model is analyzed using both analytical and computational approaches. While the model is novel and examines an important phenomenon, some parts of the manuscript do not clearly connect the math to biology. I therefore recommend that the manuscript undergo major revision.

Specific comments:

The manuscript considers a number of possible mathematical formulations for host cell-virus interaction, the immune response, and genotype mortality. There is a lack of biological justification for the mathematical form of these functions. What processes do these functions represent and why look at different functions? The section describing mathematical stability analysis of the model (section 2) is not very clear. For example, equation 2.4 suddenly introduces the variable v without explanation of how it is related to the original variable u. It is not clear how the authors arrived at the stationary solution given at the top of page 7. There seems to have been some assumption about the form of functions J(u) and S(u), but these assumptions are not stated in the manuscript. The manuscript is also not very clear on what are new results and previous work. Some specific forms of this model appear to have been analyzed in previous work, so the authors need to be clearer about what is new here (different assumptions for the mathematical form of certain interactions?). In the last sentence of section 3, what do the authors mean by efficient? 

Author Response

The authors proposed a model to study the virus evolution and diversification due to mutations, competition for host cells, and cross-reactive immune responses. The model is based on a nonlinear

partial differential equation for the virus density depending on the genotype considered as a continuous variable and on time. This model includes terms related to reaction-diffusion. In addition, the model includes nonlocal effects of virus interaction with host cells and with immune cells. In the model, a virus strain is represented by a localized solution concentrated around some given genotype. Emergence of new strains corresponds to a periodic wave propagating in the space of genotypes.

The authors described the conditions of appearance of waves (new strains) and their dynamics. The main concern in this article is the hypotheses that there could be a virus strain diffusion in the space. Please, the authors need to justify very well this strong hypothesis.

The explanation is now given after equation (1.1).

The authors considered periodic boundary conditions. The authors need to explain why from a biologically point of view. In addition, if there are periodic boundary conditions it is expected that the solution will have the periodic waves.

We have added the following comment in the discussion:

The range of genotypes is supposed to be sufficiently large in order to neglect the influence of the boundaries. Mathematically, we can consider the whole real axis. In numerical simulations, we consider a bounded sufficiently large interval. In most cases, we stop the simulations before the wave approaches the boundary of the interval. In some cases (Figure 8), we continue the simulations in order to reveal spatiotemporal pattern formation inside the interval. In this case, periodic boundary conditions are more convenient because they do not influence the behavior of the solution inside the interval. Dirichlet or Neumann boundary conditions can influence the behavior of solutions. Thus, periodic boundary conditions do not have here biological significance but they are more appropriate for mathematical modelling.

I didn’t see if the authors used a history due to the use of the delay. Some IC is needed in the interval [-tau, 0].

The explanation is now given after equation (3.3).

The authors need to improve the grammar of the paragraphs. In many places, it is not clear the ideas.

In the revised version, we have added a number of clarifying statements about the model structure, the methodology of analysis and the interpretation of the results. We hope that these extensions resolve the concerns of the reviewer.

Round 2

Reviewer 3 Report

The authors have addressed all my previous comments to my satisfaction. I support publication of the manuscript.

Author Response

Thank you.